# MicroRNAs in Acute ST Elevation Myocardial Infarction—A New Tool for Diagnosis and Prognosis: Therapeutic Implications

**DOI:** 10.3390/ijms22094799

**Published:** 2021-04-30

**Authors:** Alina Ioana Scărlătescu, Miruna Mihaela Micheu, Nicoleta-Monica Popa-Fotea, Maria Dorobanțu

**Affiliations:** 1Department of Cardiology, “Carol Davila” University of Medicine and Pharmacy, 050474 Bucharest, Romania; fotea.nicoleta@yahoo.com (N.-M.P.-F.); maria.dorobantu@gmail.com (M.D.); 2Department of Cardiology, Clinical Emergency Hospital of Bucharest, 014461 Bucharest, Romania; mirunamicheu@yahoo.com

**Keywords:** miRNA, STEMI, biomarkers

## Abstract

Despite diagnostic and therapeutic advances, coronary artery disease and especially its extreme manifestation, ST elevation myocardial infarction (STEMI), remain the leading causes of morbidity and mortality worldwide. Early and prompt diagnosis is of great importance regarding the prognosis of STEMI patients. In recent years, microRNAs (miRNAs) have emerged as promising tools involved in many pathophysiological processes in various fields, including cardiovascular diseases. In acute coronary syndromes (ACS), circulating levels of miRNAs are significantly elevated, as an indicator of cardiac damage, making them a promising marker for early diagnosis of myocardial infarction. They also have prognostic value and great potential as therapeutic targets considering their key function in gene regulation. This review aims to summarize current information about miRNAs and their role as diagnostic, prognostic and therapeutic targets in STEMI patients.

## 1. Introduction

ST elevation myocardial infarction (STEMI) remains one of the leading causes of morbidity and mortality worldwide [1]. Considering its increasing prevalence in recent years, especially in the young population, early diagnosis, accurate prognosis prediction, and rapid and efficient treatment are of paramount importance to improve patient outcome [2].

STEMI occurs due to occlusion of one or more coronary arteries, leading to transmural myocardial ischemia, which in turn results in myocardial injury or necrosis. The cause of this abrupt disruption of blood flow is usually atherosclerotic plaque rupture/erosion/fissure or dissection of coronary arteries, resulting in an obstructing thrombus. Atherosclerosis is a major contributor to coronary artery disease and myocardial infarction (MI) [3]. To this day, the initiation and progression of atherosclerosis (from the fatty streaks to vulnerable plaques that are usually responsible for the acute coronary syndromes) are not completely understood [4]. The pathogenesis of atherosclerosis is a result of interactions between multiple cell types, the vascular wall, and inflammatory factors that occur at vulnerable sites [4].

According to European Society of Cardiology guidelines, STEMI diagnosis is based on biochemical blood markers, clinical history, symptoms, ECG, and coronary angiography [1,5]. The gold standard biomarker for early diagnosis of MI is cardiac troponin I, used in clinical practice to rule in or rule out an acute coronary syndrome (ACS) [2]. However, troponin is not entirely specific for an acute coronary event; there are many other non-cardiac pathologies associated with troponin elevation (sepsis, chronic kidney injury, pulmonary embolism). Therefore there is a need for biomarkers of myocardial damage with higher sensitivity and specificity for early diagnosis, prognosis, and timely treatment of MI [2,6].

In recent years, miRNAs have emerged as promising tools involved in many pathophysiological processes in various fields, including cardiovascular diseases. miRNAs are endogenously produced small RNAs that regulate gene expression at the post-transcriptional level [7,8].

Since their discovery, circulating miRNAs have been detected in numerous body fluids (serum, plasma, saliva, tears, and urine) [9,10,11], raising interest in their potential use as markers in cardiovascular diseases. Circulating miRNAs can regulate several key cellular processes and target gene expression in recipient cells, thus conditioning cellular development, differentiation, proliferation, cell metabolism, and cell death [12]. They are being quantitatively altered in various cardiovascular diseases such as atherosclerosis, coronary artery disease—ACS/STEMI, heart failure (HF), hypertrophy, and fibrosis [13].

The miRNA biogenesis (Figure 1) starts in the nucleus where miRNAs are transcribed by RNA polymerase II to primary miRNAs (pri-miRNAs) [12]. Pri-miRNAs are processed and cleaved into precursor miRNAs (pre-miRNAs) by the RNase III enzyme Drosha and its cofactor DGCR8 (DiGeorge syndrome critical region gene) [12]. Pre-miRNAs are then exported from the nucleus to the cytoplasm by an Exportin 5-dependent mechanism and further processed in the cytoplasm by another RNase III enzyme, Dicer, producing a short double-stranded miRNA duplex [13,14,15]. The miRNA duplex is further unwound by a helicase into a mature miRNA that will be incorporated into RISC (RNA-induced silencing complex), a complex formed by the components of the Argonaute protein family members [12,16,17]. A mature miRNA can target one or more genes, while one gene can be targeted by multiple miRNAs, enabling them to be a part of a variety of physiological and pathological processes [12,13,14,18,19].

In this review, we will discuss the role of miRNAs in the pathophysiology of STEMI, their role as diagnostic and prognostic biomarkers, and last but not least, their potential as therapeutic targets in acute myocardial infarction (AMI) (Figure 2).

## 2. miRNAs in Pathophysiology of Acute Myocardial Infarction

miRNAs appear to have a role in the development and progression of ischemic heart disease at multiple levels (angiogenesis, platelet activation, atherogenesis, lipoprotein homeostasis, etc.) [20]. It is suggested that miRNAs play a critical role in the STEMI pathophysiological process as well.

Pathophysiologically, acute myocardial infarction (MI) is commonly defined as a cardiomyocyte death due to a prolonged ischaemia resulting from an acute imbalance between oxygen supply and demand [21]. Atherosclerosis, one of the major contributors to CAD and MI, is a chronic inflammatory disease of the arterial wall. The initiation and progression of atherosclerotic plaques is a multistage process with key signaling and molecular regulatory pathways [22]. Following the disruption of a vulnerable atherosclerotic plaque, it’s underlying thrombogenic core is exposed to the blood stream, resulting in thromboembolism (platelets adhere to the site of a rupture and release pro-inflammatory mediators, leading to thrombus formation) and subsequent acute coronary obstruction [23]. 

### 2.1. Atherosclerosis

miRNAs have emerged as important regulators of pathophysiological processes such as cellular adhesion, proliferation, lipid uptake, generation of inflammatory mediators. Their involvement in atherosclerotic plaque rupture/erosion is of great clinical im-portance. Plaque disruption with thrombus formation is thought to be the major patho-genetic mechanism for ACS [3]. Vulnerable atherosclerotic plaques (rupture prone) are associated with the presence of a large necrotic core covered by a thin fibrous cap, highly inflammatory cell content, and also a decreased number of smooth muscle cells (SMCs) in the fibrous cap [3]. Plaque rupture and plaque erosion are the two main plaque phenotypes in patients with STEMI. Morphologically, ruptured plaques are usually rich in lipids with macrophage infiltration, while the main components of eroded plaques are proteoglycans and SMCs [24].

Many studies demonstrated the implications of miRNAs in processes related to the risk of atherosclerotic plaque rupture/erosion. For example, miR-24 downregulation proved to enhance macrophage apoptosis, promoting plaque progression and instability [25], while miR-29 is involved in plaque stabilization by suppressing IFN-g (interferon gamma) production (IFN-g contributes to fibrous cap thinning by inhibiting the ability of SMCs to express the genes that encode procollagens) [26,27]. Overexpression of miR-145 increases stability (increased collagen content and fibrous cap area); miR-126 is linked to plaque stabilization—it increases SMCs and collagen content and reduces apoptotic cells [28]; miR-150 inhibits the formation of foam cells from macrophages exposed to the action of oxidized LDL—cholesterol [29]; therefore, it inhibits the accumulation of LDL—cholesterol molecules by macrophages and increases their release through these cells by silencing the adiponectin receptor gene [29]. miR-150 inhibits the formation of macrophage foam cells through targeting adiponectin receptor 2 [29].

Other miRNAs proved to be involved in atherosclerotic process in various studies. There was an increased expression of 5 miRNAs (miR-100, miR-127, miR-145, miR-133a, and miR-133b) in symptomatic atherosclerotic plaques compared with asymptomatic atherosclerotic plaques [30]. miR-145, miR-127, miR-100, and miR-133a/b were reported as dysregulated in plaque instability and rupture, which may result in ACS [30,31]. Li J. et al. found a set of circulating microRNAs (miR-744-3p, miR-330-3p, and miR-324-3p) associated with plaque rupture that also has the power to distinguish the plaque phenotype in STEMI patients [24], although the precise mechanisms underlying their upregulation remain to be determined [24]. In a different study, H. Dong et al proved that circulating miR-3667-3p might be a potential biomarker for distinguishing plaque erosion from plaque rupture [32]. Several various miRNAs, such as miR-210, miR-222, miR-155, miR-27a/b, and miR-221, may accompany foam cells participating in neoangiogenesis, a process that contributes to the growth of atheromas and plaque instability [33,34].

### 2.2. Thrombosis

Thrombosis also plays a critical role in the pathogenesis of ACS. Pathological platelet activation induces arterial thrombotic conditions such as MI [35]. In STEMI, the exposure and release of plaque components following plaque rupture triggers the activation of platelets. Activated platelets release microparticles that carry various cytoplasmic components, including miRNAs, thought to be involved in subsequent processes [35].

MiR-223 is one of the most abundant miRNAs in platelets. It regulates the expression of P2Y_12_, which is crucial for platelet aggregation, granule secretion and thrombus growth and stability [36]. Taking in consideration the fact that P2Y12 is also a receptor of thineopiridines (i.e.,—platelet antiagrenant Clopidogrel), miR-223 was found to be associated with platelet response to Clopidogrel in ACS patients [37].

Gidlof et al [38] observed that activated platelets also release various miRNAs (miR-22, miR-185, miR-320b, and miR-423-5p) that can regulate endothelial cell gene expression. These 4 miRNAs were increased in the supernatant of platelets after aggregation and were depleted in thrombi aspirated from AMI patients, thus proving their release from activated platelets. In the same study, ICAM-1 (intracellular adhesion molecule) expression on endothelial cells was downregulated by a miR-320b dependent mechanism [38].

### 2.3. Cardiomyocyte Death 

Acute and complete coronary artery obstruciton due to thrombosis leads to subsequent myocardial ischemia and cell death. Cell death does not occur immediately after arterial occlusion but at about 30–40 min. There are several factors that influence the time course of myocardial necrosis including the type of occlusion, collateral circulation, individual oxygen demands, and myocardial preconditioning [39]. MI involves three types of cell death: necrosis, apoptosis, and autophagy [40]. The role of miRNAs in regulating the expression of anti- and pro-apoptotic genes and regulation of cardiomyocyte and endothelial cell survival has been demonstrated in various studies [40]. Below, we provide a few relevant examples of miRNAs involved in necrosptosis and apoptosis in STEMI.

Necroptosis is a caspase-independent regulated cell death depending on receptor-interacting serine-threonine-protein kinase RIPK and p-MLKL activity [41]. In a mouse model with MI, miR-155 and miR-874 were downregulated, leading to the inhibition of cardiomyocyte death through the RIPK pathway. On the other hand, miR-103/107 and miR-2816, miR-874 were upregulated, this high expression level promoting cell death [42]. Other studies confirmed that miR-874 was able to regulate necrosis both *in vitro* and *in vivo* [43].

Apoptosis is a type of programmed cell death. Its signal is mediated by many pro- and anti-apoptotic factors. miRNAs have a regulatory role in apoptosis by targeting apoptotic related pathways [40]. A few examples: miR-320 promotes apoptosis by targeting IGF-1 (insulin growth factor 1), a growth factor that inhibits apoptosis by upregulating Bcl-2 levels and downregulating Caspase-3 levels [44]; miR-93 overexpression inhibits cardiomyocyte apoptosis by targeting phosphatase and tensin homolog (PTEN) in mice with ischemia/reperfusion injury [45]; inhibition of endogenous miR-153 can block cardiomyocyte apoptosis [46]; knockdown of miR-122, an apoptosis-related miRNA, attenuates myocardial cell apoptosis by upregulating GATA-4 [47]. Recently, it has been shown that miR-325 targets ARC (activity-dependent cytoskeleton associated protein) in the cascades of autophagy and cell death (it determines autophagy through repressing ARC, an anti-autophagic protein); overexpression of miR325-3p inhibits myocardial tissue necrosis (necroptosis) in mice, reduces the apoptotic rate, and improves cardiac function [48].

## 3. miRNAs as Biomarkers in STEMI

In cardiovascular diseases, miRNAs are discussed as potential specific biomarkers [7], involved in cell-to-cell communication, that modulate numerous signaling pathways and cellular processes [49]. Their small size, simple chemical composition, specificity, high stability, capability to withstand extreme conditions, lower complexity in comparison with proteins, and a cost-effective quantification (by RT-qPCR) make them excellent potential biomarkers [9] in acute coronary syndromes [13].

Myocardial injury leads to the appearance of cardiomyocyte-specific biomarkers in the bloodstream (i.e., circulating cardiac troponins), a phenomenon helpful for the early diagnosis of ACS. They circulate associated with miRNA binding proteins and can also be found in extracellular vesicles (apoptotic bodies, exosomes, and microvesicles) [10,11,13,50]. An early and accurate diagnosis is essential to facilitate rapid decision making and treatment (effective revascularization) and therefore improve the outcome in these high-risk patients [2].

### 3.1. Cardiac-Specific miRNAs (myomiRs)

Taking into consideration the association of miRNA with the formation and rupture of atherosclerotic plaques, their role as a diagnostic biomarker is to be expected [13]. So far, many studies have investigated the role of miRNAs in identifying patients with AMI. From over 200 miRNAs found in the heart [13,51,52], the upregulation of the same miRNAs shortly after MI [53,54] was observed: miR-1, miR-133a, miR-208a, and miR-499—classically referred to as the myomiR family (Figure 3) [55,56].

These myomiRs are abundant in the myocardium and were frequently reported as being highly increased in a myocardial infarction setting, both in animal models (first detected in mouse, pig and rat) and in STEMI patients [10,57,58,59,60]. In a recent subgroup analysis [2], miR-499 had the highest diagnostic value [61], followed by miR-133a and miR-208.

#### 3.1.1. miR-499

miR-499 is located on myosin gene (MYH7B); it can induce the structural and functional differentiation of cardiac stem cells into cardiomyocytes, thereby promoting cardiac recovery after injury [54]. It also has a cardioprotective effect by protecting the myocardium against H_2_O_2_-induced apoptosis (it targets several pro-apoptotic regulators) [62].

Much like troponin, miR-499 can be detected within a few hours after the onset of STEMI symptoms [54]. *In vivo* studies demonstrated that plasma levels of miR-499 increase 15 min after coronary ligation, almost simultaneous with troponin [63], but it takes much longer to reach a peak (24 h) compared to troponin (6 h); then they slowly decrease to baseline levels within 7 days [64]. Nevertheless, miR-499 could be detected in blood in the first 4 h after STEMI, a little earlier than troponin; therefore, it could improve the accuracy of troponin for AMI diagnosis [64]. There appeared to be a correlation between miR-499 and troponin levels, and the value of this combination proved to be superior to either of them alone, supporting the idea of its use as a biomarker [2,65].

#### 3.1.2. miR-133a

Various studies have demonstrated an elevation of miR-133 circulating levels in MI patients, indicating cardiac damage. It is a muscle-specific miRNA expressed abundantly in cardiomyocytes, playing an important role in cardiac development and hypertrophy [66]. A rapid increase in circulating miR-133a 1 h after STEMI has been observed in rats [63], with the highest levels at 3 h after symptom onset [62,63]. Its levels increase in a time-dependent manner following a trend similar to troponin I, but their levels are lower than that of troponin [63].

miR-133 also has a cardioprotective function [13] by repressing myocardial fibrosis (by directly blocking the expression of pro-fibrotic genes) and remodeling in injured myocardium [62].

#### 3.1.3. miR-208

miR-208 is a heart-specific miRNA, making it a possible diagnostic tool in STEMI. Encoded by the MYH7 gene (beta myosin heavy chain 7 gene), it regulates the expression of its host gene via the Sox6 transcription factor factor and also has a role in cardiac remodeling and fibrosis [67,68]. 

During the early stages of AMI, this miRNA might leak out of the necrotic myocardium and be released into the circulation. Low levels of miR-208 in the heart can lead to ischemia and reperfusion injury, promoting the formation of AMI [69]. Its circulating levels increased 1 hour after STEMI (undetectable in plasma before AMI), and peaked at the 3rd hour [69,70] compared with troponin that could be detected in blood at 4–8 h after the onset of myocardial injury [70]. This proves that miR-208a had greater diagnostic value during the early stage of AMI [70]. Liu X. et al. [69] revealed that miR-208, along with miR-499, displayed a more reliable value than miR-1 in AMI diagnosis. Despite this, other studies reported extremely low concentrations of miR-208a and miR-208b in AMI patients, which might make accurate detection difficult and lead to considerable errors [71].

#### 3.1.4. miR-1

miR-1 is a muscle specific miRNA highly expressed in cardiac and skeletal muscle. Appearance of miR-1 in the blood stream of STEMI patients suggest its release from necrotic myocytes [6,72]. Circulating miR-1 levels were significantly higher in STEMI patients compared to health controls in a study conducted by AI et al. [72]. The peak expression of miR-1 was similar to that of troponin I, indicating that miR-1 is a marker of cardiac damage [73]. Wang et al. [74] found a rapid increase in circulating miR-1 in rats after coronary artery ligation [75]. Circulating miR-1 peaked at 6 h of AMI onset and returned to the basal level in 3 days which showed a faster and earlier time course than known biomarkers, such as troponin [75]. Circulating miR-1 also was positively correlated with CK-MB and MI size [13]. All findings suggest that miR-1 might be a diagnostic biomarker for AMI.

### 3.2. Non-Cardiac miRNAs in STEMI

Besides the myomiRs presented above, there are many other miRNAs that could also be used as diagnostic biomarkers in STEMI [13].

Plasma miR-21 level proved to be significantly elevated in patients with AMI compared with those with angina or healthy people and has a similar diagnostic ability compared with CK, CK-MB, and troponin I levels [76]. An increased level of miR-122-5p in AMI had a high correlation with troponin I [77]. miR-221-3p was elevated in AMI patients and had a high correlation with troponin and LV (left ventricular) systolic function [78]. miR-124 was upregulated in STEMI, and had a positive correlation with troponin I and CK-MB levels. Circulating miR-124 peaked earlier than troponin [79]. miR-30d-5p had a higher diagnostic value than troponin [80]. With an upregulated expression in STEMI patients, miR-19b, miR-223 and miR-483-5p all peaked earlier that troponin in a study on 280 patients, 140 with AMI and 140 controls [81]. In another study, Wang et al. observed elevated levels of miR-122 and decreased levels of miR-22-5p in patients with AMI [82].

All these non-cardiac miRNAs mentioned above (and many others) proved to have diagnostic power in AMI but were not as favorable as the result obtained for the myomiRs. Further research is required prior to their clinical application, considering the variable results for the same miRNA.

## 4. miRNA as a Prognostic Biomarker

Studies targeting miRNAs have evaluated their prognostic value regarding two important aspects of post-myocardial infarction evolution: the ability to predict left ventricular remodeling (LVR) and cardiovascular mortality, respectively [83,84] (Table 1).

### 4.1. Left Ventricular Remodeling

LVR is a complex process involving cardiac myocyte growth and death, inflammation, fibrosis, and neovascularization [85,86]. Cardiomyocyte death occurs in the acute phase of STEMI, followed by subsequent recruitment of inflammatory cells to begin the repair process. After the resolution of the inflammatory response, fibroblast proliferation and secretion of extracellular matrix proteins leads to scar formation. In the long run, these molecular changes alter the cardiac function, leading to either beneficial or adverse LVR depending on the differences in cellular response to events [87].

In recent years, primary PCI (percutaneous coronary intervention) has significantly reduced mortality in STEMI patients by decreasing the infarct size and the extent of adverse LVR [88]. A primary reperfusion strategy with the purpose of coronary blood flow restoration is the gold standard in STEMI treatment [89]. Despite successful reperfusion, many patients develop ventricular adverse remodeling and HF after STEMI [88]. The ability to predict the risk of adverse LVR could improve early prognosis and miRNAs proved to be a useful tool in this matter. 

Several studies have reported an association between circulating miRNAs levels (miR-29a, miR-29b, miR-150, miR-30a-5p) and the development of LV dysfunction and heart failure (HF) after AMI [90,91]. A prospective study of 150 AMI patients assessed the prognostic value of a panel of 4 miRNAs (miR-16, miR-27a, miR-101, and miR-150) [92]. The plasma concentrations of these miRNAs were measured 3–4 days after STEMI along with NTproBNP values. The function of the left ventricle was assessed in all patients by echocardiography on the day of discharge and at 6 months follow-up [92]. Patients with low levels of miR-150 or miR-101 and elevated levels of miR-16 or miR-27a were at higher risk of impaired LV contractility. Measuring this combination of 4 miRNAs at discharge allowed a better prediction of post-infarction LV systolic dysfunction than the clinical prognostic indicators and NTproBNP alone. The strongest individual relationship to the development of LV systolic dysfunction in the period up to 6 months after the myocardial infarction was shown by miR-27a [92]. Among them, miR-150 has a strong individual relationship to post-infarction LVR [93]. Its cardioprotective function against myocardial fibrosis and injury, and cardiac hypertrophy was confirmed in a mouse AMI model in 2015 [94]. miR-150 upregulation in the heart led to inhibition of cardiac hypertrophy and fibrosis by regulation of serum response factor and cMyb factor and it also suppressed pro-apoptotic genes [90]. On the other hand, its downregulation was associated with hypertrophy, LV ruptures, and unfavorable LVR after STEMI [83,90]. A study led by Devaux et al. [95] found a strong negative correlation between the degree of the intensity of LVR and the plasma concentration of miR-150. Patients who developed post-infarction LVR had a plasma concentration of miR-150 that was two times lower [95]. miR-150 has a role in angiogenesis and ischemia-induced neovascularization [96]. miR-150 inhibits the expression of SRC signaling inhibitor 1 (SRCIN1), an inhibitor of Src, a tyrosine kinase necessary for VEGF (vascular endothelial growth factor) induced angiogenesis. In myocardial ischemia, the activity of SRCIN1 is considerably increased, and hence the activity of Src and endothelial NOS (eNOS) is significantly reduced. Ischemia-induced damage to human cardiomyocytes, evaluated in vitro, turned out to be significantly inhibited by miR-150 [97].

miR-155 is another miRNA related to unfavorable post-infarction LVR [98]. A lower expression of miR-155 during the second post-infarction inflammatory response phase determined the development of unfavorable post-infarction LVR in STEMI patients [98]. In a group of AMI patients studied by Zhang et al. [18], the lower plasma levels of miR-145 was significantly correlated with increased serum levels of brain natriuretic peptide and decreased ejection fraction [18].

SITAGRAMI clinical trial demonstrated that miR-1 and miR-29b had significant correlation with unfavorable post-MI LVR [99]. Grabmaier et al. measured plasma levels of miR-1, miR-21, miR-29b and miR-92a on days 4 and 9 and 6 months after AMI in a group of 44 patients and in 18 matched controls [99]. miR-1 and miR-29b levels were significantly correlated with infarct volume changes at 6 months follow-up, while only miR-29b levels were associated with changes to left ventricular end-diastolic volume over time [85,99]. Quameni et al. [85] proved that miR-1 at admission is an independent predictive factor of LVR after AMI and provides critical information for early therapeutic interventions for LVR after STEMI) [85].

Elevated plasma values of miR-208b and miR-34a can be considered predictors of the development of ventricular dysfunction and LVR after MI, associated with higher mortality at 6 months and a 23.1% higher rate of HF [100]. Lv et al. reported the elevated level of circulating miR-208b in the left ventricular (LV) remodeling after AMI and a positive correlation of miR-208b with the risk of mortality or HF [100].

miR-133a was associated with large infarcts with large areas of residual ischemia even after reperfusion, and therefore it can also be considered a predictor of post-infarction ventricular remodeling [101]. miR-133b was positively associated with microvascular obstruction and worse LV functional recovery [102].

A recent study demonstrated for the first time the role of miR-320a in predicting changes in LVEDV and occurrence of LV adverse remodeling at 6 months after STEMI in a group of 56 patients treated by primary PCI [89]. Liu et al. [103] confirmed the existence of a correlation between the expression of miR-184 and post-STEMI LVR at one month follow-up in a group of 72 patients with AMI (parameters analyzed: NTproBNP, LV end-diastolic diameter, LVEF—LV ejection fraction). Interestingly, at one year after AMI, the levels of miR-184 still varied and showed a positive correlation with MACE (major acute cardiovascular events), making it a promising prognostic biomarker [103].

### 4.2. Mortality

Regarding the ability to predict cardiovascular mortality after STEMI, many miRNAs proved to have prognostic value.

miR-133a and miR-208 were the first miRNAs identified as prognostic markers; their value at 6 months after AMI correlated with an increase in all-cause mortality [104]. The prognostic value of miR-208 was also confirmed by other studies [67,105,106]. Eitel et al. analyzed 216 consecutive STEMI patients treated by primary PCI divided into 2 groups by the miR-133a value at admission [101]. At 6 months follow-up, the primary endpoint was the occurrence of MACE (death, reinfarction, HF). Elevated miR-133a concentration was associated with less myocardial salvage, larger infarct area, and left ventricular dysfunction at MRI [101]. Elevated miR-208b expression was associated with reduced long-term survival in AMI [67].

Another miRNA that proved to be effective in predicting mortality after STEMI (on follow-up after 30 days, 4 months, and 1, 2, and 6 years, respectively) is miR-499 [106,107]. Increased levels of miR-208b and miR-499-5p were strongly associated with increased risk of mortality or heart failure within 30 days after AMI [105]. Also, elevated circulating levels of miR-134, miR-328 and miR-145 are associated with increased mortality and development of HF after AMI [104,108,109].

Matosumo et al. analyzed a group of 4160 patients with STEMI and observed higher levels of miR-155 and miR-380 in patients who experienced cardiac death within one year after discharge; miR-192, miR-194, and miR-34 were significantly higher in the serum of patients who later developed heart failure [110,111,112]. Another prospective study of 1002 patients with STEMI demonstrated that miR-320a, miR-26b-5p, and miR-660-5p are associated with major cardiovascular events within 1 year of follow-up and with an increased risk prediction when added to the GRACE score [113]. 

Cortez-Dias et al. proposed a ratio of miRNAs as a prognostic marker. They evaluated the value of the miR-122-5p/133b ratio at the time of catheterization as an early prognostic marker in AMI [114]. In a group of 142 STEMI patients, the ones with unfavorable outcome (all-cause mortality, death or MI, any adverse cardiovascular event) had higher miR-122-5p/133b ratios at the time of the urgent cardiac catheterization, had lower LVEF, and were more likely to present with left main or multivessel coronary disease [114].

## 5. Therapeutic Potential of miRNAs in STEMI

In addition to their efficiency in both diagnosis and prognosis, miRNAs have recently emerged as promising therapeutic targets in STEMI [86,115]. So far, miRNA therapeutic modulation techniques were used in the settings of atherosclerosis, acute myocardial infarction, vascular remodeling, arrhythmias, hypertrophy and fibrosis, angiogenesis, ischemic injury, etc. [116,117,118].

In STEMI, miRNA therapies have different cell types as targets: cardiomyocytes, inflammatory cells, fibroblasts, and endothelial cells [115]. Two approaches are used in most studies to target cardiomyocytes after AMI: prevention of cardiomyocyte death after ischemia and induction of cardiomyocyte proliferation after resolution of ischemic injury [115]. Fibroblast activation after AMI plays an important role in scar formation, essential to prevent ventricular wall rupture; however, excessive cardiac fibrosis can lead to a pathological response, exacerbating cardiac injury and subsequent HF [115]. Angiogenesis is another important process that helps restore blood flow to the infarcted tissue, essential for cardiac repair after AMI [115]. Regarding the inflammatory response, novel therapies should aim to dampen the initial inflammatory response that can further harm the cardiac tissue and promote the anti-inflammatory reparative phase, favoring healing and scar formation, thereby limiting infarct size [115].

Difficulties in the use of therapeutically altering miRNAs lie in the fact that a single miRNA can affect the expression of numerous cell types and genes, while at the same time miRNA-based regulation can involve a vast number of different miRNAs [119]. Therefore, miRNAs have a relatively modest effect on their target [7,119].

Treatment options involving miRNAs aim to specifically alter the levels of miRNAs: suppression of miRNAs/raising miRNA levels or substituting them by artificially generated copies, depending on the pathophysiologic mechanism [119]. This can be achieved by miRNA mimic technology. This implies to generate nonnatural double-stranded miRNA-like RNA fragments that bind specifically to its target mRNA and induce gene suppression. As opposed to endogenous miRNAs, miR mimics function gene specifically [7,120]. miRNA suppression can be obtained by antagomirs [121], a class of chemically engineered oligonucleotides specifically silencing single endogenous miRNAs [122]. They competitively inhibit specific miRNAs by binding to the target mature miRNA and lead to a reduced activation of RISC and consequently to an upregulation of specific mRNAs and gene expression [7,123].

So far there have been 213 studies regarding 116 different miRNAs [115]. A few of the most relevant in vivo studies that hold potential for patient use in the future are detailed below.

miR-26a attenuated cardiac ischemia/reperfusion injury, inflammatory cell infiltration, and cardiomyocyte apoptosis [124,125,126]. In a mouse model of MI, the expression of miR-26a was significantly decreased in the infarcted zone of the heart, whereas apoptosis and ATM (ataxia telangiectasia mutated) expression were increased [127]. The overexpression of miR-26a decreased cardiac apoptosis and fibrosis through repression of ATM, and it was beneficial for the reduction of cardiac dysfunction [127]. It also improved cardiac function and reduced cardiac fibrosis by lowering the expression of collagen type I and connective tissue growth factor (CTGF) in mice 2 weeks after MI [127,128].

Lesizza et al. aimed to assess the functional effect of a single-dose, intramyocardial injection of synthetic miRNA mimics after myocardial infarction in mice [129]. After permanent ligation of the left anterior descending artery, an injection with miR-199a-3p and miR-590-3p mimics was administered in the infarct border zone. LVEF was preserved in the animals treated with miR-590-3p throughout the whole echocardiographic follow-up period (up to 8 weeks after AMI), as opposed to controls. In miR-199a-3p-treated mice, LVEF was similar to controls. Highly significant reduction of the infarct size was observed after 8 weeks in both miR-199a-3p and miR-590-3p groups [129]. A single administration of miR-199a-3p and miR-590-3p mimics preserves cardiac function and leads to cardiac repair after STEMI [129]. Another recent study demonstrated that the expression of human miR-199a in infarcted pig hearts can stimulate cardiac repair [130]. It is known that cardiomyocyte proliferation is under the control of miRNA [115]. One month after the induction of MI, miRNA was administered in pigs through an adeno-associated viral vector [130]. By stimulating cardiomyocyte de-differentiation and proliferation, Gabisonia and colleagues observed diminished cardiac fibrosis, increased muscle mass and reduced scar size, and marked improvements in both global and regional contractility [130]. However, long-term expression of this miRNA eventually resulted in sudden cardiac death of most test subjects (70% of treated animals) [115]. Therefore, further testing and dose adjustment is required [131].

miR-221 inhibits ischemia-induced apoptosis both *in vitro* and *in vivo*. The anti-apoptotic effect of miR-221 is reflected by the downregulation of the pro-apoptotic genes Puma, Bmf, and Bak1 in cardiac myocytes [132,133]. Reduced miR-221 expression is associated with severe cardiac fibrosis in HF patients. miR-221 increased in the infarcted and peri-infarct areas 2 days after AMI in rats treated with miR-221 mimics after left coronary artery ligation [132]. miR-221 mimics enhanced cardiomyocyte survival by reducing apoptosis and autophagy. This led to reduced infarct size and cardiac fibrosis, with improved cardiac contractility and less adverse remodeling on days 7 and 30 in the rat MI model [132].

miR-133a is known to be useful for enhancing the regenerative properties and survival of transplanted stem cells and cardiac progenitor cells (CPCs), and for reprogramming mature non-cardiac cells to cardiomyocytes—most transplanted mesenchymal stem cells (MSCs) undergo cell apoptosis in the ischemic myocardium microenvironment [134]. Overexpression of miR-133a protects CPCs from cell death by targeting the pro-apoptotic genes Bim and Bmf, thus improving cardiomyocyte proliferation after AMI [135]. This was demonstrated by Chen et al. in 2017 when using a rat MI model; they observed that transplantation of miR-133-overexpressing MSCs improved cardiac function after AMI [136]. Inflammation and infarct size decreased in miR-133-MSC-injected rat hearts by repression of snail 1 by miR-133-overexpressing MSCs [136]. Consequently, miR-133 could be an effective target to promote MSC survival in the ischemic myocardium microenvironment [134].

miR-92a is upregulated in cardiac myocytes after MI and it is transferred to cardiac fibroblasts within exosomes. In fibroblasts, miR-92a relieves the SMAD7-mediated inhibition of αSMA transcription, triggering the conversion to myofibroblast. Overexpression of miR-92a contributed to the activation of fibroblasts [137]. miR-92 is also involved in angiogenesis. Studies showed that it targets ITGF5 (a proangiogenic integrin alpha 5) and that its inhibition substantially increases angiogenesis and granulation tissue formation [138]. In an MI mouse model, systemic administration of an antagomir against miR-92a on days 0, 2, 4, 7, and 9 after AMI resulted in functional recovery of damaged tissue (improved heart function and reduced infarction size after 14 days) by enhanced blood vessel growth, reduced apoptosis, and improved MI size [139].

miR-21 has a strong anti-apoptotic effect on vascular SMCs and cardiomyocytes. Mi-21 levels are reduced in the infarcted area of rats after MI but upregulated in border areas. Dong and colleagues (2009) were able to inhibit the downregulation miR-21 expression in the infarcted areas by ischemic preconditioning, thus leading to increased levels of miR21 [139]. Overexpression of miR-21 via Adenovirus-mediated miR-21 gene transfer decreased cell apoptosis, decreased myocardial infarct size by 29% at 24 h, and also decreased LV dimensions 2 weeks after AMI, thus improving LVR [139].

Another miRNA-based therapy technique that holds great potential for treatment of myocardial infarction targets cardiac macrophages. Bejerano et al. [140] studied the effect of nanoparticle-based targeted delivery of miR-21 mimic to cardiac macrophages. They demonstrated that boosting miR-21 expression in cardiac macrophages at the infarct site during the first days after MI accelerates the switch of inflammatory macrophages to the reparative state, leading to increased angiogenesis, lower number of apoptotic cells, and attenuation in left ventricle remodeling after MI [140]. Localized injection of miR-21-enriched extracellular vesicles reduces AMI-associated cell apoptosis, increases the number of viable cardiomyocytes, and reduces scar formation, thus improving cardiac function after AMI [141].

Both miR-21 and miR-146a have anti-apoptotic effects (by inhibition of caspase 3 pathways) and therefore beneficial effects on ischemic injury [142,143,144]. In a study from 2016 [145] miR-21 and miR-146a synergically decreased apoptosis under ischemia/hypoxic conditions in mice cardiomyocytes compared with either miR-21 or miR-146a alone. Mice injected with agomiR-21 and agomiR-146a had decreased infarct size and increased ejection fraction compared to miRNAs applied individually [145]. In conclusion, this combination of antagomirs attenuates cardiac dysfunction and apoptosis after AMI [145].

miR-320, another potential therapeutic target for myocardial ischemia, has a role in cell proliferation, myocardial ischemia/reperfusion injury (I/R injury) and stimulates cardiomyocyte death and apoptosis [146,147]. miR-320 expression worsens myocardial I/R injury, by directly inhibiting the IGF-1 and preventing IGF receptor-mediated activation of the PI3k/AKT pathway, while its inhibition protects against myocardial apoptosis [146]. Using a lentivirus expressing miR-320 inhibitor restored the IGF-1 function and determined a decrease in the number of apoptotic cardiomyocytes and preserved cardiac function [147]. miR-320 inhibition using antagomir-320 protects the left ventricle from remodeling after myocardial I/R injury [147]. The knockdown of miR-320 protects rat cardiomyocytes against I/R injury by upregulation of heat shock protein 20 (a cardioprotective molecule) [143,144]. A decrease in the infarction size was observed after the administration of miR-320 antagomiR [148].

miR-29 targets ECM (extracellular matrix) protein mRNAs, including collagens, fibrillin, and elastin and appears to be decreased after AMI [149]. Downregulation of miR-29 with anti-miRs in vitro and in vivo induces the expression of collagen, whereas overexpression of miR-29 in fibroblasts reduces collagen expression, concluding that miR-29 acts as a regulator of cardiac fibrosis and represents a potential therapeutic target for tissue fibrosis [149]. By using an miR-29 antisense inhibitor in a mouse ischemia-reperfusion model, another study group was able to protect cardiomyocytes from injury [149,150]. The authors even succeeded in reducing infarction size with antagomirs against miR-29a and miR-29c [150].

Downregulation of miR-122 reduces hypoxia/reoxygenation-induced myocardial cell apoptosis via upregulation of GATA-4 [47]. Liang et al. [47] demonstrated using a hypoxia/reoxygenation model of rat cardiomyocytes H9C2 in vitro that miR-122 is upregulated in hypoxic myocardial cells. GATA-4, one of the most important cardiac transcription factors that participates in myocyte proliferation and survival [151] is a direct target gene of miR-122 and has a protective role against hypoxia-induced cardiomyocyte injury [152]. Its expression is inhibited by miR-122 upregulation and upregulated by miR-122 inhibition [47]. Overexpression of miR-122 by recombinant adeno-associated viral vector infection markedly promoted the apoptosis of H9C2 cells, whereas miR-122 inhibition significantly decreased cell apoptosis [47]. Hence miR-122 may serve as a promising target for the prevention of myocardial H/R injury [47].

All these studies provide evidence of the use of miRNAs as a therapeutic target in AMI, considering their potential to improve the outcome of STEMI patients. However, the main problem with miRNA therapy is that a single miRNA controls the expression of many genes and changing the expression of an miRNA can cause various side effects [93]. Tissue-specific delivery methods could be a solution for this issue [93]. In the light of the recent clinical trial of an antisense drug targeting miR-132 in HF patients that showed promising results [153], clinical trials with miRNAs in STEMI are eagerly awaited in the near future.

## 6. Conclusions

miRNAs have emerged as a key epigenetic mechanism in cardiovascular diseases. Their good accessibility, high sensibility and specificity, and feasible methods of detection make them suitable for use in clinical practice [13]. However, larger multicenter trials are required to establish whether they actually offer additional benefits over the existing diagnostic and prognostic biomarkers in STEMI. In vivo studies developed so far also support the potential use of miRNAs as therapeutic targets, but further human studies are required until their current use in clinical practice.

Although there are several limitations to be resolved until their clinical use, it is certain that circulating miRNAs show great potential in diagnosis and prognosis and also as therapeutic targets in STEMI.

## Figures and Tables

**Figure 1 ijms-22-04799-f001:**
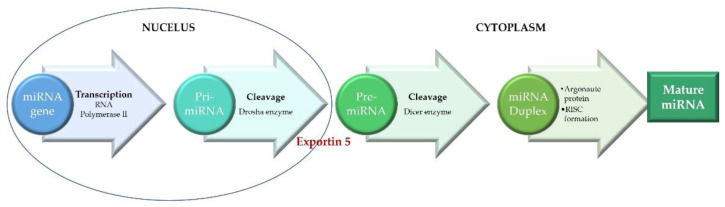
miRNA biogenesis.

**Figure 2 ijms-22-04799-f002:**
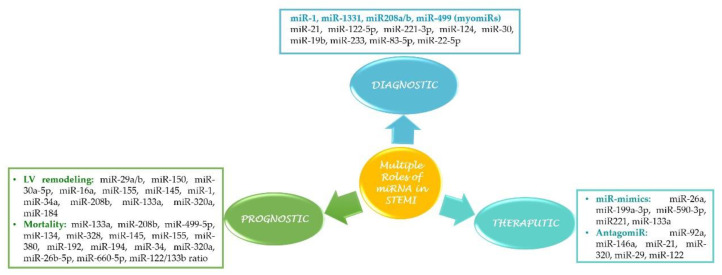
Multiple roles of miRNA in STEMI.

**Figure 3 ijms-22-04799-f003:**
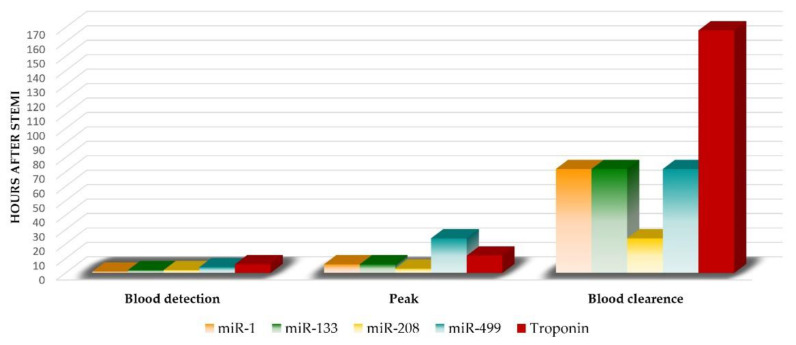
Comparative kinetics of circulant miRNA and troponin in STEMI.

**Table 1 ijms-22-04799-t001:** miRNA as a prognostic biomarker.

miRNA Prognostic Value (LVR and Mortality after STEMI)		
	Favorable	Unfavorable
miRNAs	miR-150 (LVR) miR-145 (LVR) miR-101 (LVR)	miR-27a, miR-16 (LVR) miR-155 (LVR and mortality) miR-1, miR-29b (LVR) miR-208b (LVR and mortality) miR-34 (LVR) miR-133a (LVR and mortality), miR-133b (LVR) miR-320 (LVR and mortality) miR-184 (LVR) miR-499 (mortality) miR-134, miR-328 (mortality) miR-380 (mortality) miR-192, miR-194, and miR-34 (LVR) miR-26b, miR-660 (mortality) miR122-5p/133b ratio (mortality)

## Data Availability

Not applicable.

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
