# Peer review of "MicroRNAs in Acute ST Elevation Myocardial Infarction—A New Tool for Diagnosis and Prognosis: Therapeutic Implications"

_ijms, 2021, doi:10.3390/ijms22094799_

Round 1
Reviewer 1 Report
This is an excellent review about the role of MicroRNAs in acute ST elevation myocardial infarction
Authors provided a comprehensive overview of microRNAs as a new tool for diagnosis, prognosis and therapeutic implications. Number of overviewed references are more than sufficient.
I have a minor suggestion I would like to suggest creating a table for subsection No.4.It should be practical divide microRNAs into two groups on the basis of good or bad prognostic role regarding remodelling and mortality.
Reviewer 2 Report
The authors aims to review the area and relevance of miRNA as prognostic markers of STEMI. This is highly relevant to the field and should be of interest to many researchers both clinical and basic.
However, the overall readability and flow of the paper is poor. For example, the authors do not provide sufficient information and motivation to why they study STEMI. Very sparse information on the STEMI definition, which is key and should be presented upfront in the introduction. Instead, the authors provide bits and pieces of information to describe pathophysiology of STEMI all over the paper. The authors should put STEMI in a larger context and provide a clear definition before they discuss the literature. The introduction part is crucial to and need to be re-constructed in order to understand the following section where they address atherosclerosis and plaque rupture, the build-up of foam cells, thrombosis and arterial occlusion.
The authors should re-structure the paper completely. Move section that starts at row 39 to the beginning to increase readability. The same stands for section that start on row 55 … and many more.
Since the author would like to review the link between STEMI pathophysiology and the relevance of mRNA markers. The definition of STEMI and its complexity should be highlighted and explained before addressing specific miRNA
References issues:
The author should not referee to other review articles when summarizing findings in the literature. For example, on row 81-82, the mentioned mRNAs in in relation to plaque instability are not original paper but a review. It is a danger when a review reviews other reviews and not the actual literature. This is the same for reference 22.
In reference 19 the authors refer to a paper with 11 case and 11 controls, which is a clearly underpowered study to investigate prognostic value of specific miRNAs
The annotations need to be consistent and clearly defined:
Row 250 LVR is defined but the LV is used on row 252.
AMI is not defined
IGF is first mentioned at row 122 but then defined at row 488. Similar is found for many others. It seems like the paper have been re-assembled and it makes it very hard to read.
